# Aspects of Self-Management After Solid Organ Transplantation—A Scoping Review

**DOI:** 10.3390/nursrep15080304

**Published:** 2025-08-19

**Authors:** Stefan Jobst, Christiane Kugler, Anne Rebafka

**Affiliations:** Institute of Nursing Science, Faculty of Medicine, University of Freiburg, Breisacher Str. 153, 79110 Freiburg, Germany; stefan.jobst@uniklinik-freiburg.de (S.J.);

**Keywords:** solid organ transplantation, self-management, conceptualization, scoping review

## Abstract

**Background**: Solid organ transplantation improves survival and quality of life but requires lifelong self-management. While models exist for kidney and liver recipients, a comprehensive framework for all solid organ transplant recipients is lacking. Addressing this gap is essential for optimizing post-transplant care. **Objectives**: This report aims to conceptualize self-management after solid organ transplantation by addressing questions related to (1) the contexts studied to date, (2) research methodologies and publication types used, and (3) core aspects associated with self-management post-transplantation. **Methods**: A scoping review was used to address the above objectives. A comprehensive search strategy identified relevant studies, followed by systematic screening, data extraction, and qualitative content analysis. Findings were categorized using a deductive–inductive coding approach to map core self-management aspects after solid organ transplantation. **Results**: The search yielded 34,417 records, with 742 ultimately included. Publications from 43 countries spanned 43 years, with many (48.9%) published after 2016. Research articles dominated (80.1%), covering kidney (61%), liver (22%), heart (21%), and lung (16%) transplants. A qualitative analysis identified four self-management domains containing various categories: (1) Managing the medical–therapeutic regimen, (2) managing biographical work, (3) managing (new) life roles, and (4) generic self-management skills. The conceptual model illustrates their interconnections, with aspects of the medical–therapeutic regimen management most frequently covered. **Conclusions**: Self-management after solid organ transplantation is complex, involving medical–therapeutic, emotional, social, and behavioral aspects. Aspects of managing the medical–therapeutic regimen dominate the research literature, while other aspects need further exploration. Future studies should address gaps to support holistic, patient-centered post-transplant care strategies.

## 1. Introduction

For individuals suffering from end-stage organ diseases, transplantation of the affected organ may be the best possible therapy [1]. Solid organ transplantation (SOTx) comprises heart, lung, kidney, liver, pancreas and small intestine [2]. SOTx is a complex but long-established and cost-effective procedure that saves lives and has been shown to improve the quality of life and social functioning of transplant recipients [3,4,5,6]. While SOTx is a treatment for a life-threatening illness, recipients remain chronically ill which is accompanied by lifelong medical–therapeutic treatment and lifestyle changes to prevent complications such as rejection or infections, or to mitigate the side effects of drug treatment [7,8,9,10,11].

The medical–therapeutic treatment regimen after SOTx includes (1) drug therapy to prevent graft rejection (immunosuppression), infections and metabolic diseases (e.g., hypertension or diabetes), (2) monitoring of graft function, and (3) preventive strategies, i.e., avoiding risk factors and carrying out health-promoting activities [12]. Implementing this complex treatment regimen, overcoming challenges, and maintaining health and well-being involves numerous tasks and can only be achieved through the interaction of healthcare providers and transplant recipients and their relatives. This requires the active involvement and self-management of transplant recipients in their daily life after transplantation [9,13,14,15].

Self-management encompasses everything that people with a chronic condition do in their daily lives to cope with the effects of their condition [16]. Self-management is a mature, socially constructed concept that is often described in the context of chronic illness [17,18,19,20]. It is embedded in the paradigm shift towards a more collaborative model of healthcare, which anticipates active involvement of patients [21,22]. This corresponds more closely to the reality of life for (chronically) ill people, who are embedded in a system of professional healthcare, but who can only perceive a small part of the illness experience and the associated symptoms and problems of those affected [23,24]. Day-to-day management of the condition is performed by patients, which underscores the importance of self-management and necessary competencies [18,25]. In view of the life-changing impact of transplantation and the above-mentioned tasks within the post-transplant regimen, self-management is essential for transplant recipients, as reflected in many studies [9,14,15,26,27,28,29,30,31,32,33,34,35], and positively affects, among other things, treatment adherence [32,36,37,38], health-promoting behaviors [32,39], graft function [39,40], and psychosocial aspects [28,33,39,41,42].

The most comprehensive conceptualization of self-management after organ transplantation to date is available for kidney transplant recipients and was presented by Schäfer-Keller et al. in a conceptual model [43]. This model is based on research findings on the self-management of chronically ill patients [44,45] and was transferred to the specific needs of kidney transplant recipients. The tasks of self-management after a kidney transplant are divided into three areas: The management of (1) medical–therapeutic regimen, (2) emotions and (3) (new) life roles. These areas are embedded in and influenced by contextual factors and people who support the patient’s self-management. These factors include the organ transplant recipients themselves, their families/relatives and professional healthcare providers, as well as other organ transplant recipients [43]. With regard to the self-management of recipients of other organs, only two studies were identified that summarized self-management after liver transplantation at a conceptual level [32,46].

While organ transplant recipients do have specific self-management requirements depending on the transplanted organ, it is reasonable to conclude that there are more similarities than differences at the meso-level. This assumption is further supported by the findings of van Zanten et al. [47], which indicate that facilitating self-management is crucial for all transplant recipients, irrespective of the transplanted organ. Consequently, it can be inferred that a shared foundation for self-management exists among all organ transplant recipients. A comprehensive overview covering the concept of self-management after organ transplantation and including all organ entities could not be identified in the literature. This lack of a summarizing overview of self-management after organ transplantation at a meso-level risks overlooking important unmet support needs and requirements of organ transplant recipients, which in turn compromises a more holistic approach to self-management support. Therefore, a better understanding of the concept of self-management after SOTx is needed to provide a comprehensive evidence base for healthcare providers and researchers to better support adult organ transplant recipients in their self-management. The Schäfer-Keller model can serve as a reference point for this endeavor [48].

## 2. Aims

The overarching aims of this scoping review (ScR), as outlined in a previously published protocol [49], were (1) to explore and map existing evidence on self-management and self-management support in adults after SOTx, and (2) to identify aspects, challenges, and/or needs related to life after SOTx that are amenable to be addressed by self-management. This report focuses on the first aim and the conceptualization of self-management after SOTx by addressing the following questions:In which contexts (temporal, geographical) has the concept of self-management after SOTx been studied to date?Which research methodologies were used in the eligible evidence and which publication types are provided?What were the core aspects associated with self-management after SOTx?

Another review question concerning the definition and/or operationalization of the concept of self-management following SOTx [49] has already been addressed in another publication [50].

## 3. Materials and Methods

The reporting of this ScR is structured in accordance with the guidelines set forth in the Preferred Reporting Items for Systematic reviews and Meta-Analyses extension for Scoping Reviews (PRISMA-ScR) [51].

### 3.1. Design

An ScR is a type of evidence synthesis that aims to systematically identify and map the breadth of available evidence on a specific topic, area, concept or problem. This is often carried out independently of the source (i.e., primary research, reviews, non-empirical evidence) and within or across a specific context. ScR can clarify key concepts and/or definitions in the literature and identify key characteristics or factors associated with a concept, including those related to methodological research [52] or analyzing knowledge gaps [52,53]. This broad and comprehensive approach was chosen as literature on self-management is often difficult to identify and analyze due to inconsistent or synonymous use of terminology and various definitions [54].

### 3.2. Eligibility Criteria

This review considered evidence relating to adults following human SOTx (heart, lung, liver, pancreas, kidney, or small bowel transplantation, including combined forms) that focuses on their self-management and associated needs and challenges. Eligibility criteria in the Concept domain were based on definitions of self-management by Wilkinson & Whitehead [55], the classification of “everyday problems” [20], and self-management support by Orrego et al. [56]. Included evidence sources were published evidence (primary studies, evidence syntheses, ongoing studies/study protocols, letters to the editor(s), conference papers) of all designs and research methods in English or German, irrespective of country or time of publication. The detailed list of inclusion and exclusion criteria has been published elsewhere [49].

### 3.3. Search Strategy and Information Sources

A comprehensive and sensitive search strategy was developed in collaboration with a team of experienced librarians and an information retrieval specialist. First, a search of Medline (Pubmed) and Google Scholar, using key terms (“self-manage*” and “solid organ transplant*”), identified relevant evidence that was analyzed by the research team to extract these key terms. In parallel, by reviewing and analyzing models, concept analyses, and definitions of the concept of “self-management” [16,18,19,45,57,58,59,60,61,62,63,64,65,66,67,68,69,70] further keywords were identified. This iterative process led to three conceptual building blocks of terminology related to “organ transplantation”, “self-management” and the “organ transplant/healthcare provider perspective” which was then operationalized for an initial search in Medline (Ovid). All identified keywords and medical subject headings were collected, merged and discussed within research group meetings. The resulting search strategy was reviewed, validated and finalized by the information retrieval specialist using the Peer Review of Electronic Search Strategies guideline [71]. The final search strategy for Medline (OVID) [49] was then adapted to search the following electronic databases: CINAHL (EBSCO), PsycINFO (EBSCO), Emcare (OVID), Web of Science (Clarivate), and the Cochrane Library (Wiley). In addition, ClinicalTrials.gov (www.clinicaltrials.gov), the WHO International Clinical Trials Registry Platform (www.who.int), and the German Clinical Trials Registry (www.drks.de) were searched for ongoing or completed and unpublished trials. The respective search strategies are provided as Appendix A. The search was supplemented by recommendations from experts and hand searches. The searches in the databases and trial registers were conducted at the end of September 2021. Manual searches were performed repeatedly from the start of the screening process until April 2022. The time period to be searched on the respective databases was not limited due to the explorative nature of this ScR and its purpose to discover the breadth of the literature of the evidence concerning self-management after SOTx [53,72].

### 3.4. Selection of Sources of Evidence

After removing duplicates using the reference management tool EndNote (Version X8) [73], five reviewers performed screening and selection of relevant documents in a two-stage process based on the eligibility criteria using Covidence [74]. First, the titles and abstracts of the deduplicated documents were screened and irrelevant references were excluded. Subsequently, the full texts of all potentially relevant documents were obtained and reviewed. In both stages of the screening process, each document was reviewed independently by two reviewers and any discrepancies in the assessments were discussed. If no consensus could be reached, the primary investigator was consulted to make a final decision.

The screening process was tested in advance by all reviewers as part of a pilot phase with 25 randomly selected documents. Insights were discussed among all reviewers, any discrepancies were resolved, and necessary adjustments or clarifications were made and then recorded in an internal manual on the selection and screening process.

### 3.5. Data Extraction Process

The extraction of data from included documents involved five reviewers and was performed using a standardized form developed by the authors (S.J.; A.R.) based on the questions of the ScR and the recommendations of Peters et al. [75] and Pollock et al. [76]. The data extraction form was embedded in Covidence [74] and was used to extract (1) general/bibliographical information, (2) methodological/methodological information, (3) information on the population, and (4) specific information relating to the questions of the ScR (aspects, challenges/needs, definitions). All parts of included documents (with the exception of the bibliography and organizational information [e.g., ethics, acknowledgments, etc.]) were considered for data extraction. Relevant data were extracted in either text or numerical form. Text was extracted verbatim. Deviations from the original wording (omissions, added words) were marked with square brackets. Each included full text was extracted independently by two reviewers and finally consented to by a member of the research team with expertise in self-management and organ transplantation (S.J. or A.R.). Any ambiguities or discrepancies were discussed and resolved.

The data extraction process was initially tested in a pilot phase. Each reviewer extracted four methodologically different full texts to ensure the functionality and clarity of content of the data extraction form. The test phase was debriefed with all reviewers and minimal changes were made to the functionality of the form on Covidence.

### 3.6. Critical Appraisal

In line with current methodological recommendations, no critical appraisal was carried out as the purpose of this ScR was to map available evidence and not to provide a consolidated and clinically relevant answer to a question or to make a statement about the effectiveness of interventions [53,76].

### 3.7. Data Analysis and Synthesis of Results

Prior to the analysis, the extracted data were checked and processed. First, all data were converted into a tabular format (Microsoft Excel 2016 [77]), checked for errors, and amended by hand if necessary. This table was then divided according to the main topics of “Definitions/Operationalization”, “Aspects”, and “Problems, Needs/Requirements”, while retaining the relevant document characteristics, thus allowing separate analyses. Quantitative and qualitative analyses were carried out in accordance with the objectives and questions of the ScR.

For the mapping of existing evidence, bibliographic data and characteristics of included documents (such as year, country, document type, methodology, number of participants, organ type) were analyzed quantitatively by applying descriptive statistics [53,75,76] using Microsoft Excel 2016 (Version 16) [77]. Findings were presented in tabular and graphical form.

Since this report focuses on the analysis of the aspects of self-management after organ transplantation, only the method for analyzing the data extracted for this purpose is described below. Corresponding data were available in unprocessed form, comprising sentences, partial sentences, and word combinations. Therefore, as recommended by Pollock et al. [76] and Morse [78], these data were analyzed using qualitative content analysis with a deductive–inductive approach [79,80], carried out in three phases [79,81]. In a preparation phase, a deductive coding frame was developed, the structure of which was based on an analytical framework design derived from the best-fit framework synthesis methodology [82]. This coding frame integrated the core elements of models/concepts of self-management following kidney [43,83,84] or liver transplantation [32]. The resulting initial deductive coding frame was validated by an experienced qualitative researcher. During the organizing phase, manifest data were firstly coded and categorized using de- and re-contextualization using the software MAXQDA 2022 (Version 22.4.1) [85]. The extracted texts of included documents were each defined as a unit of analysis within which meaning units (constellations of words or statements that refer to the same central meaning and contain aspects that are linked by their content and context) were identified [86]. A “condensation” of meaning units was not carried out, as published scientific texts can be considered “condensed” per se in terms of content and language [87]. Secondly, codes were deductively assigned to the categories of the previously developed coding frame. Codes that did not match were then inductively combined into new categories. These inductive categories were integrated into the initially developed deductive coding frame, which was then revised and adapted by combining, moving or renaming categories without changing the basic structure of four domains. By counting the number of documents with content on the respective categories, their frequency was calculated and thus a quantification within the category system was determined. In the reporting phase, a narrative description of the coding frame and its contents was carried out, of which a graphical representation in the form of a model was generated.

## 4. Results

The literature search resulted in a total of 34,417 records, of which 11,323 were removed as duplicates. By screening the titles and abstracts of the remaining 23,094 records, 21,690 were excluded and 1404 documents were included in the full-text screening. Of these, 29 documents were not retrievable and 593 documents were excluded. Finally, 782 documents were included. These included 40 documents (register entries, study protocols), which could also be assigned to included research reports and merged with 33 documents. As a result, 742 data records were ultimately included in the data analysis, encompassing a publication period from 1979 up to 2022. Figure 1 shows the Preferred Reporting Items for Systematic reviews and Meta-Analyses (PRISMA) flow diagram for the screening and selection process [88]. The corresponding checklist is provided as Appendix A. References for the documents included in this ScR are available in Appendix A.

### 4.1. Characteristics of Sources of Evidence

The period of publication of the included documents covers 43 years. The earliest publication dates from 1979, the most recent publications from 2022. The median year of publication is 2015, with an interquartile range of 11 years (1st quartile = 2008; 3rd quartile = 2019). Almost half of the included documents (n = 353; 48.9%) were published between 2016 and 2022, with most documents published in 2020 (n = 70) (see Figure 2).

In total, documents from 43 different countries were included. In terms of geographical regions of the world as defined by the United Nations Standard Country or Area Codes for Statistical Use [89], 75% of the included documents (n = 554) originate from Europe and North America (Figure 3). The vast majority of the included documents were written in the form of research articles (n = 594; 80.1%), followed by (expert) opinion/discussion articles (n = 66; 8.9%) and best practice articles (n = 48; 6.5%). For research articles, study protocols and study register entries (n = 625; 84%), the respective methodological approach reported was recorded. Descriptive/explorative (quantitative), qualitative and experimental (quantitative) methods were used most frequently (Table 1), comprising 557 articles that included a sample of participating organ transplant recipients. The total number of participants analyzed was 90,662, with a median of 74 (range: 1–3467).

Forty-five articles declared as evidence syntheses and three mixed methods articles that integrated an evidence synthesis using a systematic approach were identified. Forty of these articles contained precise information on the number of studies included. A total of 1119 studies were included in these evidence syntheses, ranging from 3 to 147 and a median of 19.5 studies.

Almost two thirds of the included documents (n = 455; 61%) focused on kidney transplantation, followed by liver (n = 164; 22%), heart (n = 154; 21%) and lung transplantation (n = 122; 16%). Only a few documents addressed aspects of self-management after pancreas or small bowel transplantation (n = 41; 6%) (Table 2).

### 4.2. Conceptual Model of Aspects of Self-Management Following Organ Transplantation

Using qualitative content analysis, 4374 text passages from the 742 included documents were coded and summarized in a hierarchical coding frame into four domains—(1) Managing the medical–therapeutic treatment regimen, (2) managing biographical work, (3) managing (new) life roles, and (4) generic self-management skills, strategies and processes—and 9 categories and 27 sub-categories. A comprehensive table detailing all category levels can be found in Appendix A. The final version of the coding frame represents the foundation of the conceptual model of the self-management of adults following SOTx (see Figure 4). The model is characterized by a circular structure, with the domains 1–3 at its center. Within the four domains, the associated categories are shown, which are in turn linked to associated aspects (subcategories). The overlap of the domains illustrates the partial connections between individual aspects. These are framed by overarching self-management skills, strategies and processes (domain 4), which may influence all aspects. All domains and categories will be summarized narratively below. The number of documents containing codes for each category is given for each category level. Appendix A provides a complete overview of which documents informed which categories and subcategories.

#### 4.2.1. Managing the Medical–Therapeutic Regimen (n = 683)

This domain includes self-management aspects of the entire medical–therapeutic treatment process after transplantation, described within two associated categories.

Managing key elements of the therapy (n = 575): Key elements of therapy to be managed by SOTx recipients comprise the management of medication, self-monitoring of indicators of health, managing symptoms and side effects, keeping regular appointments with healthcare providers, understanding transplantation and its therapy, building a partnership with healthcare providers, and caring for surgical wounds or drains.

Managing one’s health related lifestyle (n = 448): Following SOTx, a healthy lifestyle, i.e., the (daily) performance of health promoting and illness preventing behaviors, can help to avoid complications such as infections, or of metabolic or cardiovascular diseases. This includes eating healthily and appropriately, exercising/physical activity, preventing or controlling infections, refraining from use of harmful substances, preventing skin cancer, and managing body weight. These aspects are related to learning or implementing appropriate, transplant-related and lifelong self-care practices of recipients as part of activities of daily living in order to maintain the best possible health for themselves and the donated organ.

#### 4.2.2. Managing Biographical Work (n = 201)

This domain describes inner processes for redefining identity and life goals by means of two categories.

Experiencing and managing emotions (n = 162): Although no subcategories emerged, a wide range of emotions was identified as being experienced by organ transplant recipients in the period following transplantation. The management of these emotions is described as dealing with and adapting to the emotional burdens and fluctuating highs and lows that result from the challenges of life after SOTx. Most notably, coping with stress after transplantation involves managing physical, emotional, and social challenges, such as side effects, rejection fears, and anxieties about organ loss or death. Further emotional aspects include dealing with feelings of guilt, grief, or responsibility toward the donor. Additionally, uncertainty about the future and health outcomes can lead to frustration, disappointment, sadness, or depression. Effective coping relies on various skills and strategies to address these complex emotions.

Managing the self-concept (n = 96): After an organ transplant, adjustment, improvement or reconstruction of one’s identity and self-concept, including psychological and physical aspects, is often required to achieve a sense of coherence and well-being encompassing the integration of the new organ, managing reduced energy and physical capacity, and dealing with changes in appearance.

#### 4.2.3. Managing (New) Life Roles (n = 227)

This domain encompasses the adaptation or reorganization of life roles following SOTx, described within three categories.

Managing changes in social fabric (n = 168): One aspect of life roles is the integration of organ transplant recipients into a social structure, with a particular focus on social interaction. Following SOTx, recipients have more opportunities to rejoin their social environments, which often involves psychosocial adjustment or reintegration. This includes seeking support from others, maintaining and adapting relationships, and managing intimate relationships.

Performing everyday activities (n = 84): Life following SOTx encompasses the execution, adaptation, and management of everyday tasks, which are reflected in various aspects, inextricably linked to “Managing one’s health related lifestyle”. Occupation and returning to work requires early planning, employer agreements, and specialist advice. In some cases, it may involve career changes or adjustments to reduce occupational stress and health risks. Leisure activities like travelling or gardening provide balance but require planning to minimize risks. Household tasks and mobility, such as driving or public transport, demand habit changes and infection prevention measures. It is also crucial to manage finances effectively, with recipients addressing financial constraints and treatment costs to avoid undue burdens.

Adjusting to (a new) normality or reality (n = 82): Organ transplant recipients wish to and require a return to a state of normality. However, this process frequently necessitates the adaptation to a “new” normality that requires accepting and integrating challenges (e.g., complications) into daily life. This adaptation is a physical, mental, social, and habitual process, requiring behavioral adjustments and psychological integration of the transplant experience. Recipients must accept that they may never attain a state of complete health or a conventional notion of normality. They should adapt to their new health reality resulting from the transplant, for example by reconciling past expectations with their current and future realities. The goal is to achieve a sense of normality by embracing life as it is.

#### 4.2.4. Generic Self-Management Skills, Strategies, and Processes (n = 240)

This fourth domain comprises skills, strategies, and processes within two categories that can be regarded as superordinate and linked to aspects of the domains described above or influence the implementation of the tasks and activities mentioned therein.

Self-management skills (n = 133): The subcategories “taking action” and “health literacy” summarize self-management skills. Taking action in self-management after SOTx requires active engagement, responsibility, problem solving, and decision-making. Recipients should take a proactive role in recovery and maintaining a healthy lifestyle, collaborating with healthcare providers. This involves responsibility for one’s health and self-management. Problem-solving and decision-making skills are key, helping recipients identify issues, barriers, and opportunities to find solutions. These abilities apply to both disease-related challenges and everyday problems, enabling recipients to actively participate in decisions that benefit their health and well-being. Health literacy is a set of skills organ transplant recipients should develop to recognize, process, and apply complex health information in daily life. Willingness to (lifelong) learning is crucial and effective communication skills are key for exchanging information, especially with healthcare providers.

Strategies and processes in the context of self-management (n = 159): In order to cope with (psychological and physical) challenges and stressors in life following SOTx, the development and application of various internal, self-regulating strategies and processes can be helpful for organ transplant recipients. Both strategies and processes refer to the way an action or task is performed and the underlying mental and cognitive factors that shape its execution. Strategies and processes that were identified in the documents include acceptance, self-efficacy and resilience, hope and optimism, setting goals and priorities and developing plans, spirituality, and adopting a new mind-set.

### 4.3. Content Coverage of the Category System According to Organ Entity and Publication Date

#### 4.3.1. Organ Entity

Figure 5 illustrates the content coverage in relation to the domains (data on small bowel/intestinal transplantation were not included in the analysis due to an insufficient number of documents). The content on the management of the medical–therapeutic regimen is most frequently represented for all organ entities with a share of 87–100%. The other domains are each represented proportionally with around one third.

#### 4.3.2. Publication Decade

Figure 6 shows the content coverage in relation to the domains. This initially shows an even distribution of the proportion of documents with content on the four domains in the 1980s. Since the 1990s, however, an increased and consistently high proportion of documents with content on “Management of the medical–therapeutic regimen” can be observed.

## 5. Discussion

To explore existing evidence on the self-management of adults following organ transplantation, an ScR was conducted, including 742 documents from 1979–2022. A descriptive analysis of the characteristics of the included documents revealed the breadth and depth of the available evidence. Qualitative content analysis of the documents was used to identify aspects of self-management after organ transplantation and to develop a conceptual model. The type and scope of the identified evidence, as well as the structure and content of the conceptualization, will be discussed below, followed by a discussion of individual aspects.

### 5.1. Type and Extent of Included Evidence on Self-Management Following Solid Organ Transplantation

This ScR can be described as a “large scoping review” as it comprises a high number of included documents and shows a high level of complexity due to its purpose, the larger amount of data to be processed and the more complex analysis methods [90]. A large ScR can make a valuable contribution to the understanding of broad evidence in a particular area by allowing an entire body of evidence to be captured and characterized and thus identifying future research priorities [90]. To achieve the objectives of this ScR, it was essential to include and deeply analyze a large amount of evidence using a sensitive search strategy. Limiting the number of documents or utilizing a purely descriptive analysis would have significantly reduced its informative value.

The ScR covers a large publication period of 43 years. The number of publications in the first 30 years is comparatively low in relation to the marked increase that has been observed since 2008, particularly from 2016 onwards. This reflects the growing attention paid to the topic, influenced by several factors. On one hand, the notion of “self-management” for (chronic) illness first emerged in the late 1960s, was subsequently formalized in the 1980s, and has since gained prominence [16,45]. On the other hand, SOTx became established in the 1980s due to advances in surgical techniques, effective drugs (e.g., cyclosporine), organizational structures (e.g., establishment of the UNOS database in 1982) and legal regulations [3,91,92,93,94,95]. These developments led to more transplants, longer survival rates, and a greater focus on self-management for transplant recipients [92,93,96].

The majority of included documents originate from North America and Europe, reflecting the dominance of these regions in health research due to established research infrastructures and funding opportunities [97,98,99,100]. Consequently, the self-management aspects identified in this review are largely shaped by a European–North American perspective. The representation of organ types in the review aligns with the overall transplant distribution of the past two decades [101,102,103,104].

### 5.2. Structure of the Conceptualization

The overarching structure of the newly developed conceptualization is similar to that of Corbin and Strauss [44] and Lorig and Holman [45]. At the same time, it integrates and expands the underlying conceptual areas of self-management after kidney transplantation, as developed by Schäfer-Keller et al. [43], and shows specific elements and aspects after organ transplantation of all solid organs. Here, some categories show links or overlapping content, so that they are not completely mutually exclusive. However, this interconnectedness reflects the complexity of this reality [87].

The most extensive domain is “Managing the medical–therapeutic regimen”, for which content could be identified in over 90% of the included documents. In this respect, the management of medical–therapeutic aspects can be interpreted as the main focus of self-management. Medical–therapeutic therapy and follow-up care after organ transplantation are complex and time-consuming [12,104,105,106]. They serve to maintain graft function and detect and prevent complications at an early stage and aim for the patient’s fullest possible reintegration into normal life. In addition to medical and therapeutic elements, this also includes preventive and health-promoting components. All ten “tasks” or “central components” mentioned in the Schäfer-Keller model [43] are integrated into the medical–therapeutic regimen in the present conceptualization. The spectrum of aspects has been expanded by the integration of “understanding transplantation and its therapy”, “caring for surgical wounds/drains” and “managing body weight”. The aspect of “building a partnership with healthcare providers”, classified as a skill by Schäfer-Keller and other conceptualizations, was interpreted as a task in the present conceptualization. The closely related skill of communication, which can be regarded as a fundamental prerequisite for a successful partnership with healthcare providers [25,107,108], was conceptualized as a skill as part of health literacy.

The other three domains were addressed less frequently, with content identified in only about one third of analyzed documents. This trend persisted across individual entities and the last 40 years of publications. Notably, “Managing biographical work” and “Managing (new) life roles” appear to have been less researched, even though isolated early works suggest emotional/psychological implications of organ transplantation (e.g., [109]). This has already been observed for kidney transplant recipients by other researchers [43,110] who have recommended further conceptual research. A qualitative systematic review by Rebafka et al. [26] on aspects of self-management after lung transplantation also calls for more research on psychosocial and emotional aspects beyond medical–therapeutic management. Likewise, a recent meta-synthesis found that interventions predominantly target medical–therapeutic management, emphasizing the need for high-quality studies promoting daily management and participation in end-stage renal disease patients [111]. One possible explanation for this may be the prevailing view in the professional community that “the transplantation process is more concerned with functional capacities and return to the activities of daily living than the subjective experience of individuals” [112].

In many documents on self-management after organ transplantation, the management of emotions and life roles are mentioned or considered as areas alongside the management of the medical–therapeutic regimen with reference to the work of Corbin and Strauss [44] and Lorig and Holman [45], but are rarely conceptualized in detail [9,27,34,46,113,114]. Research indicates that organ transplant recipients are sometimes confronted with serious emotional, identity-related, and psychosocial problems and challenges and have a need for support in this regard [15,27,46,110,112,115,116,117,118]. Our conceptualization of self-management after organ transplantation has highlighted relevant aspects more clearly and thus closes gaps identified by the Schäfer-Keller model [43]. In this context, the “emotional management” frequently mentioned in the literature was summarized alongside the management of the self-concept within “Managing biographical work”.

The aspect domain “Generic self-management skills, strategies and processes” integrates skills, strategies and processes on a theoretical level that are linked to all aspects of all other aspect domains. Initially based on the five self-management skills by Lorig and Holman [45], widely referenced in studies on chronic illness [119] and organ transplantation [43], this domain was later reorganized and expanded during the qualitative content analysis. Although the terms ‘skills,’ ‘strategies,’ and ‘processes’ are often used interchangeably in self-management literature, their distinction in the present concept is reasonable. Skills are defined as specific abilities with practical and cognitive components used to perform actions or tasks [70,120]. In contrast, processes and strategies refer to how tasks are executed or represent underlying cognitive factors influencing their performance [70]. While this distinction may appear somewhat artificial and there may be some overlap, it has enabled a more diverse presentation of content within this aspect domain. This diversity and complexity is also reflected in other studies on self-management of chronic illness. Schulman-Green et al. [70] identified over 70 skills across 12 processes, while the TEDSS framework describes 26 strategies in seven areas [121]. Although the present conceptualization identifies fewer elements, it shows substantial alignment with these works in content despite differing classifications. This underlines the importance of skills, strategies, and processes for self-management, particularly in the actual implementation of tasks or the performance of activities in one’s own experiential context (“self-tailoring”), which would be less likely without self-management skills [45]. Nevertheless, in the present conceptualization, these “generic” skills and processes after organ transplantation were only addressed in about one third of the documents analyzed.

### 5.3. Reflections on the Individual Aspects of the Conceptualization

#### 5.3.1. Managing the Medical–Therapeutic Regimen

This category includes managing therapy and a health-related lifestyle. Core aspects involve self-monitoring, managing symptoms and side effects, attending follow-ups (ideally with healthcare provider collaboration), and acquiring treatment knowledge. The key focus, however, is on medication management, with particular reference to lifelong immunosuppression as one of the fundamental pillars of therapy after organ transplantation [122], requiring adherence, sourcing, storage, and understanding of drug action. However, non-adherence remains a persistent challenge [123], with rates ranging from 1% to 55%, contingent on the transplanted organ, and an estimated 25% of recipients are considered non-adherent [124,125,126,127]. Non-adherence carries a risk of transplant loss, emphasizing the necessity for support and targeted interventions supporting medication self-management, as numerous interventions have shown [128,129].

The management of a health-related lifestyle is of paramount importance after organ transplantation. Common aspects include the consumption of a balanced diet, the undertaking of regular physical activity, and the effective management of body weight. These behaviors have been demonstrated to positively affect physical and mental health, mitigate chronic diseases, and reduce cardiovascular risks, particularly relevant for recipients prone to metabolic changes [130,131,132]. However, the evidence regarding optimal dietary recommendations post-transplant remains limited [133], and physical activity guidelines primarily focus on moderate to vigorous training 3 to 5 times weekly, with excessive training posing health risks [131,134]. Changing diet and exercise behavior is complex and challenging and adherence to diet and exercise recommendations remains low (19–25%) [127], indicating a need for long-term individualized support [37].

Another critical aspect within the conceptualization is the prevention and control of infections, which are a leading cause of morbidity and mortality post-transplant, requiring prevention strategies [135,136]. Although the rate of infections decreases after the first year after transplantation, in line with the dose reduction of immunosuppression [105,137], the risk remains elevated for life [135]. Surprisingly, the number of documents addressing this aspect is significantly lower than the number of documents addressing the aforementioned aspects relating a health-related lifestyle. This discrepancy can be attributed, at least in part, to the perception that infection prevention, particularly during the heightened risk period of the first year post-transplant, is predominantly the domain of healthcare providers in inpatient settings [135,136,137,138]. Despite the paucity of research in this area, it is vital to recognize the importance of infection prevention measures in the post-transplant context, and greater emphasis on education and the integration of preventive strategies outside the hospital setting is therefore essential [135].

#### 5.3.2. Managing Biographical Work

The management of biographical work addresses the psycho-emotional aspects of self-management following organ transplantation. Emotions are integral to the illness experience, acting as reactions to, expressions of, or parts of the adaptation process [139,140]. This conceptualization identified a range of negative emotions experienced by transplant recipients, including stress, anxiety, guilt, sadness, and depressive moods. While this list is not exhaustive due to the individual experience and the complexity of human emotion, which is reflected in different taxonomies or categorizations for structuring emotions [141], it aligns with the emotions referenced in the extant literature within the context of the self-management of chronic conditions, suggesting their relevance [142,143]. The prevalence of affective disorders, including depression and anxiety (20–60%, depending on the organ), underscores the necessity for addressing emotional challenges following transplantation [118]. These disorders are associated with elevated mortality rates [117], emphasizing the importance of emotional support to enhance self-management skills and prevent mental health complications [110,118,144,145].

However, the focus on rather negatively connoted emotions in the present conceptualization does not imply a pessimistic view on life after transplantation. The experience of negative emotions fluctuates over time after organ transplantation and tends to characterize the early postoperative phase and the later years of life expectancy [146]. Rather, positive emotions, such as happiness and gratitude, are also common, as transplantation is often viewed as a gift or second chance [15,28,145,146]. Self-management, however, is understood as problem-oriented [45], which is why only emotions that are generally interpreted as problems are considered in the present conceptualization. Nevertheless, positive emotions can be used as a resource, for example in the context of supportive interventions [147,148].

Emotional processing following organ transplantation is closely linked to one’s self-concept, which often requires adaptation after transplantation [149,150]. For recipients, the experience of continuing to live despite an otherwise certain death can be associated with complex feelings related not only to themselves and the deceased donor but also to the continuation of the donor person in their own self [151]. Integrating the transplanted organ into one’s own body self can be challenging, involving emotional ambivalence between gratitude and guilt or grief [118,144,145]. An organ transplant may cause experiences of alienation relating to both the recipient’s own body and the idea of the foreignness of the donated organ [152]. Reports of personality changes post-transplantation indicate different effects on different areas of personality, which may be perceived both positively and negatively [115,152,153,154]. While the empirical basis for this phenomenon remains limited, current research suggests a subjectively perceived change in many affected individuals [115]. This underscores the need for psychosocial support and the necessity for additional research on this special area of self-management post-transplantation.

#### 5.3.3. Managing (New) Life Roles

The management of (new) life roles is centered on the integration of organ transplant recipients into social structures, thereby facilitating social participation and the establishment of a “new” normality and emphasizes the social implications of chronic health conditions [155,156]. Social interaction is of paramount importance, functioning as both a resource for support and a foundation for the development of personal relationships. The establishment of proactive relationships fosters the development of social networks, thereby providing support that enhances quality of life and mitigates psychological complications, such as PTSD [157,158]. Peers and self-help groups have been shown to play a crucial role in this regard [159,160], offering emotional and practical support that strengthens self-management and disease acceptance of chronically ill patients [161,162,163,164]. Interacting with their peers enables organ transplant recipients to draw on the experiences of others, offering them emotional support and a sense of belonging, which can contribute to better physical, psychological, and social adjustment [27,34,165]. Furthermore, resuming professional activity is another central aspect, considered a key indicator of social reintegration and quality of life [4,155]. However, return-to-work rates vary greatly and a significant proportion of organ transplant recipients do not appear to pursue professional activity after transplantation [166,167]. It is therefore recommended that support for professional reintegration should begin as early as the pre-transplant [168,169]. Research has shown that returning to normal life is often achieved in the first year post-transplant and is expressed through social and occupational participation, as well as leisure activities [4]. This “new” normality is indicative of an adaptation to unforeseen events, thereby reducing susceptibility to stress [170,171].

#### 5.3.4. Generic Self-Management Skills, Strategies and Processes

This domain was organized differently compared with other conceptualizations of self-management of the chronically ill, by integrating strategies and processes in addition to skills. Here, “taking action” and “health literacy” represent self-management skills. “Taking action” includes proactive action, problem-solving and decision-making skills. Health literacy is becoming increasingly important due to the rapid progress of information and communication technologies [172,173].

Health literacy in particular is considered to be of great importance in the context of organ transplantation, with significant effects on self-management. In their study, Chisholm-Burns et al. (2018) [174] found that the ability of organ transplant recipients to understand and apply health information has a significant impact on the post-operative course and long-term treatment outcomes. Low health literacy is associated with poorer treatment outcomes, such as higher serum creatinine levels, more frequent hospitalizations, increased mortality and a higher risk of graft failure. It seems necessary to consider health literacy as an integral part of post-organ transplant care in order to strengthen the self-management of those affected and ensure long-term graft function. Health literacy is also seen as an active process that is influenced by personal and contextual factors. Affected individuals gradually develop a hierarchy of information sources that they prioritize according to reliability and relevance [175]. Even if health literacy can improve over time after transplantation [40], supportive interventions are still recommended as part of aftercare [175].

Acceptance, resilience, and self-efficacy emerged as key processes in self-management after organ transplantation. Although identified in only a small percentage of analyzed documents, acceptance appears to be an integral mechanism facilitating adaptation to post-transplantation circumstances [70,121]. It serves as a mediator for emotional and psychological adjustment, contributing to improved quality of life and effective self-management [176,177]. A partial acceptance of a health problem, however, may lead to an unstable and vulnerable lifestyle that can be disrupted by stressors, thus significantly complicating self-management [119,170]. Similarly, resilience supports self-management of chronic illness by aiding adaptation and well-being, correlating negatively with distress and impairment while being positively associated with quality of life and health promoting behavior [178,179]. Despite its associations with lower psychological distress and better health outcomes [180,181], research on resilience in transplant recipients remains limited. Self-efficacy is a critical factor in self-management [182], recognized as both a prerequisite and an outcome [67,119], influencing health behaviors and overall well-being [61]. Self-efficacy helps to maintain good mental and physical health and has a positive effect on the quality of life of organ transplant recipients by strengthening problem-solving strategies, partnership with healthcare providers, a healthy lifestyle and self-care behavior [183,184,185].

Given the impact of these processes and strategies, targeted interventions—such as psychological counselling, resilience training and structured support programs—should be implemented to strengthen self-management and ultimately improve post-transplant adaptation and long-term health outcomes [31].

### 5.4. Strengths and Limitations

This paper presents the first comprehensive conceptualization of adult self-management after transplantation of any organ. The conceptualization was conducted through an extensive, guideline-led scoping review using a sensitive search strategy that identified and analyzed a large body of research literature. Qualitative content analysis was used to illustrate not only the breadth but also the depth of the identified literature as part of the conceptualization.

However, this review has some limitations. First, the large scope should be emphasized, which led to complex and time-consuming analyses [90]. This high effort delayed the dissemination of the results and limited their topicality, but at the same time increased the informative value of the study, as a broad spectrum of research literature could be analyzed in depth. In addition, only published research literature was considered, while grey literature and other sources, such as websites, were not included in the analysis. In addition, despite the sensitive search strategy, it cannot be assumed that all publications relevant to the topic were identified. Another limitation of this review may be that 61% of documents—and thus all of the associated information—relate to kidney transplantation, which is the most extensively researched and performed solid transplantation. Therefore, domains pertaining to aspects relevant to other solid organ transplantations, especially pancreatic and small bowel transplant recipients, may not have been included due to the scarcity of research in these fields.

## 6. Conclusions

By analyzing 742 documents from a publication period of 43 years, a conceptual framework was developed, providing valuable insights into the self-management of adult organ transplant recipients. Twenty-seven aspects of self-management were identified, which were assigned to nine categories and four domains. The results support, integrate and extend the underlying conceptual domains of self-management as elaborated by Schäfer-Keller et al. [43] and identify specific elements within each domain.

The overarching structure of the conceptualization demonstrates that self-management after organ transplantation is comparable to that of chronic diseases in many areas. At the same time, it reflects the complexity and multifaceted nature of self-management after organ transplantation, highlighting its extensive and multifaceted nature.

This work confirms the finding of previous research—that the overall research literature to date has focused heavily on the management of the medical–therapeutic regimen, while other areas, such as biographical work and the management of new life roles, have received less attention. Therefore, future research should focus more on these underrepresented areas in order to obtain a more complete picture of self-management after organ transplantation.

Although some aspects of self-management were only identified in relatively few documents in the quantitative evaluation of the content analysis results, this does not necessarily imply a statement about their importance. Rather, this could also be related to the data evaluated, which reflect the perspective of researchers rather than necessarily that of those affected, i.e., organ transplant recipients and their relatives.

The findings of this ScR can help to deepen the understanding of self-management after organ transplantation and thus contribute to improving the care of organ transplant recipients. However, the resulting conceptualization does not represent a guide for practitioners, but rather offers clues and insights for supporting the self-management of their patients and can be used as a theoretical framework for the development and implementation of evidence-based interventions.

## Figures and Tables

**Figure 1 nursrep-15-00304-f001:**
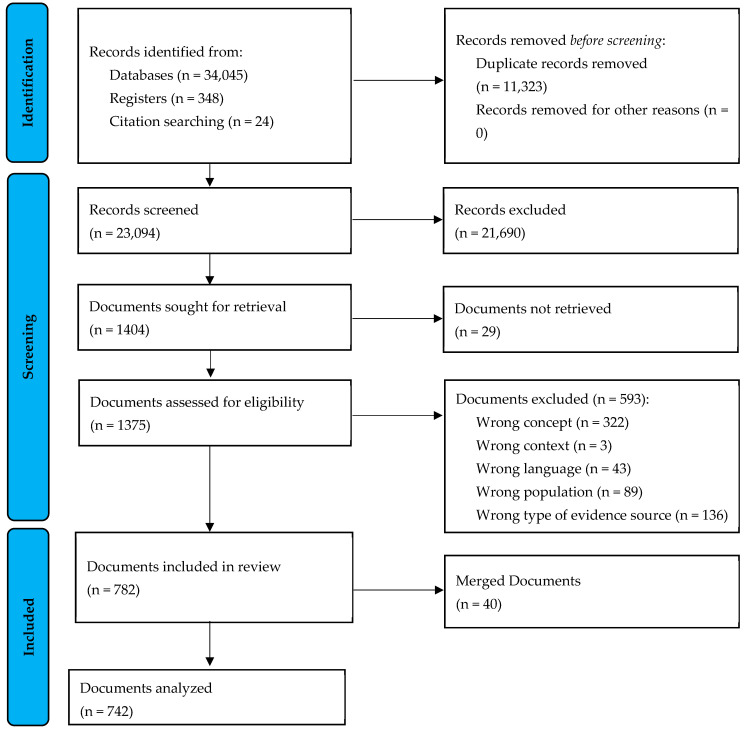
Preferred Reporting Items for Systematic reviews and Meta-Analyses (PRISMA) flow diagram for the screening and selection process in the Scoping Review.

**Figure 2 nursrep-15-00304-f002:**
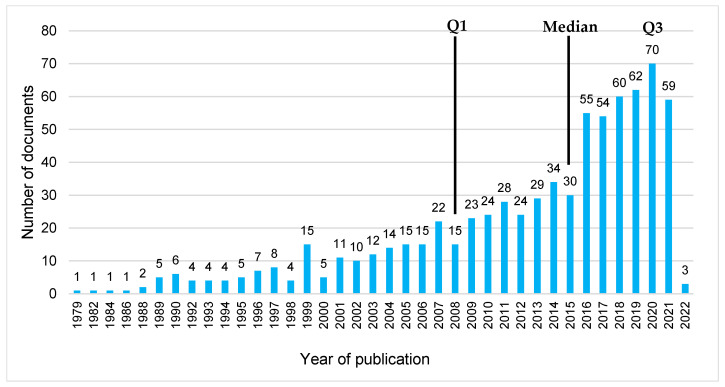
Number of included documents included in the ScR by year of publication (N = 742). Legend: Q1: 1st quartile; Q3: 3rd quartile.

**Figure 3 nursrep-15-00304-f003:**
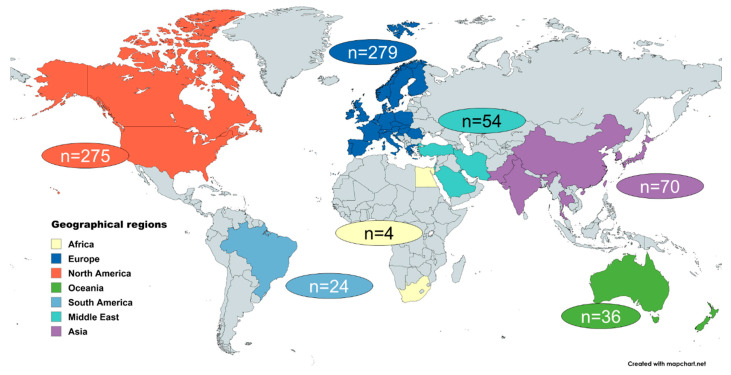
Number of included documents by origin (world region) (N = 742). Legend: Created with MapChart.net.

**Figure 4 nursrep-15-00304-f004:**
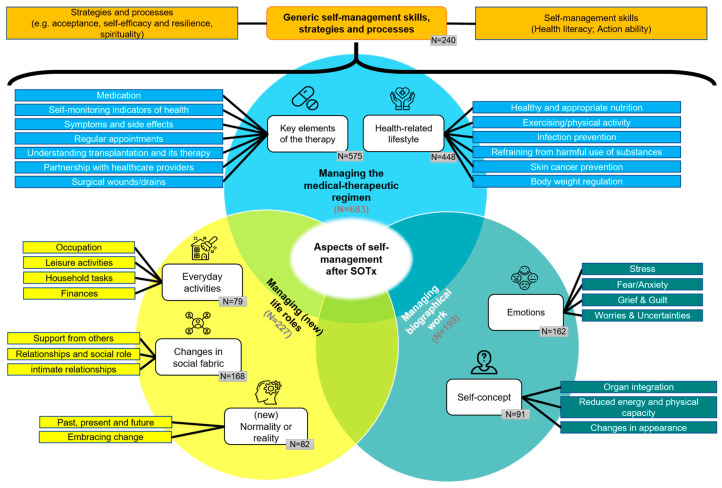
Conceptual model of self-management of adults following solid organ transplantation. Legend: N indicates the number of documents with corresponding content; icon graphics were obtained from Freepik, fzyn, Kiranshastry, Smashicons and srip via www.flaticon.com.

**Figure 5 nursrep-15-00304-f005:**
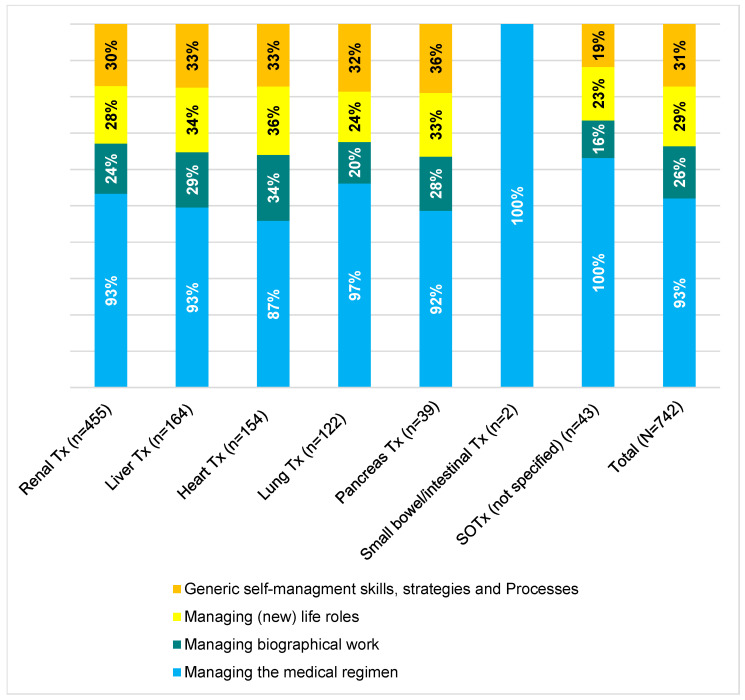
Content coverage of the category system according to organ entity and publication date. Legend: Tx: Transplantation; SOTx: Solid organ transplantation.

**Figure 6 nursrep-15-00304-f006:**
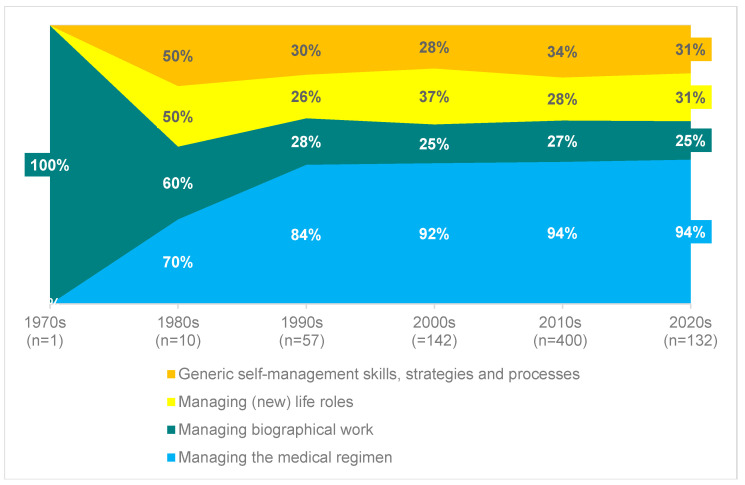
Content coverage in relation to the main domains.

**Table 1 nursrep-15-00304-t001:** Reported methodology in included research articles, study protocols and study register entries.

Methodology	Number	%
Descriptive/explorative (quantitative)	283	45.3%
Qualitative	130	20.8%
Experimental (quantitative)	106	17.0%
Mixed methods	46	7.4%
Evidence synthesis	45	7.2%
Secondary data analysis	12	1.9%
n.a. *	3	0.5%
Total	625	100%

* Three study register entries gave no indication of the methodological approach.

**Table 2 nursrep-15-00304-t002:** Overview of the organ transplant populations addressed in the included documents.

	Number of Documents	% (of 742)	% (of 979)
Kidney	455	61.3%	46.5%
Liver	164	22.1%	16.8%
Heart	154	20.8%	15.7%
Lung	122	16.4%	12.5%
Pancreas	39	5.3%	4.0%
Small bowel/intestine	2	0.3%	0.2%
Solid organs (not specified)	43	5.8%	4.4%
Total	979 *	132%	100%

* Some documents reported on more than one transplant type.

## Data Availability

The datasets used and/or analyzed during the current study are available from the corresponding author on reasonable request.

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
