# Peer review of "Aspects of Self-Management After Solid Organ Transplantation—A Scoping Review"

_nursrep, 2025, doi:10.3390/nursrep15080304_

Round 1

Reviewer 1 Report

Comments and Suggestions for Authors

Dear Authors, 

I wish to congratulate you to this scoping review which is very much wanted and appreciated. Again professor Kugler has manifested her excellent scientific skills. 

It is great that the you already at line 41 state the important message that SOTx is a chronic condition. A strength is that you include all types of solid organs despite the fact that you have managed to exclude 22 publications from the Scandinavian Self-Management after Thoracic Transplantation (SMATT) project along with other recent studies from e.g. Denmark. It is difficult to judge if it is a conscious bias since the keywords in those publications well match your search strategy.

The explorative nature of the ScR and its purpose to discover the breath of the evidence concerning  self-management is also strength as well as the screening process tested in advance as a part of a pilot phase.

The rigour of the method is very well described and easy to follow. The flow diagram gives a clear view of the screening process and selection process. The majority of the documents were published 2020 which guarantees new knowledge.

The number of participants included in the review are > 90 000 representing mainly a North American and European perspective.

In conclusion, the informative value of this review is very high despite the exclusion of a large amount of recent Scandinavian research particularly within the scope of this review.

Reviewer 2 Report

Comments and Suggestions for Authors

This scoping review on self-management after solid organ transplantation is a very large review of literature in this area.

The background is helpful, however it would be stronger if there was a rationale for why one meso-level model for summarizing the self-management in solid organ transplant is needed. There are some significant differences in the various types of solid organ transplant so why is a single model important in this area?

The methods are described well in the referred articles. It is a bit hard to understand the review without reading the two related articles for the methods protocol and concept review.

The search strategies and analyses are described in sufficient detail.  Results are described well. It would be helpful to have the actual references in the various areas noted in some way when the number are noted for the various domains and categories. While it would not be possible to include the references in the text, could they be noted in the reference list in some way?

Under the limitations, it should be noted that 61% of the documents relate to kidney transplantation so the predominant information is related to kidney transplantation and there may be missing domains or categories related to other types of transplantation which have not been researched, especially for pancreatic and small bowel transplant patients.

Reviewer 3 Report

Comments and Suggestions for Authors

Many thanks for this manuscript which comprehensively describes a scoping review. I do not have a lot of comments since the methodology was described in detail (and the topic is not my area of expertise). However I would encourage the authors to provide some citations in the results section, i.e. refer to 'data' (papers) where relevant to support their themes/ concepts identified. I appreciate that this is challenging given the number of papers included in the review but maybe some e.g. could be included.

Reviewer 4 Report

Comments and Suggestions for Authors
  • Please inform the study design and study location in the title of the study
  • The urgency of the study was lacking. The study’s novelty was also lacking. Please improve the introduction significantly.
  • The study design requires improvement to achieve higher internal validity, as the current design does not align well with the stated research objectives

Round 2

Reviewer 4 Report

Comments and Suggestions for Authors
  • Please inform the study design and study location in the title of the study
  • The conclusion part of the abstract section did not represent the result. It seems more complete than the result section. Please improve the abstract section to be in harmony.
  • The manuscript presents self-management as a central issue; however, the urgency and impact of this issue are not convincingly established. The rationale for focusing specifically on self-management over other potentially more critical areas—such as monitoring relevant biomarkers—remains unclear. The background section should be strengthened by providing a more compelling justification for why self-management is a priority concern in patients with solid organ transplantation. This includes discussing its relevance, potential outcomes, and supporting evidence from current literature.
  • The final set of papers to be analyzed must clearly specify the publication period in the method section, indicating the range of years (e.g., from [year] to [year]) during which the included articles were published.
